# Development and Implementation of an Internet Survey to Assess Community Health in the Face of a Health Crisis: Data from the Pregnancy and Birth Survey of the Fukushima Health Management Survey, 2016

**DOI:** 10.3390/ijerph16111946

**Published:** 2019-06-01

**Authors:** Hironori Nakano, Kayoko Ishii, Aya Goto, Seiji Yasumura, Tetsuya Ohira, Keiya Fujimori

**Affiliations:** 1Radiation Medical Science Center for the Fukushima Health Management Survey, Fukushima Medical University, Fukushima 960-1295, Japan; kayokoi@fmu.ac.jp (K.I.); agoto@fmu.ac.jp (A.G.); yasumura@fmu.ac.jp (S.Y.); teoohira@fmu.ac.jp (T.O.); fujimori@fmu.ac.jp (K.F.); 2Department of Epidemiology, Fukushima Medical University, Fukushima 960-1295, Japan; 3Center for Integrated Science and Humanities, Fukushima Medical University, Fukushima 960-1295, Japan; 4Department of Public Health, Fukushima Medical University, Fukushima 960-1295, Japan; 5Department of Obstetrics and Gynecology, Fukushima Medical University, Fukushima 960-1295, Japan

**Keywords:** Fukushima nuclear accident, pregnancy, surveys and questionnaires, internet

## Abstract

The Pregnancy and Birth Survey of the Fukushima Health Management Survey is a questionnaire survey that has been conducted annually since 2011 in Fukushima Prefecture. Since 2016, the survey has been available online as well as in paper form. This study aimed to determine whether making the survey available online improved response rates and to identify the characteristics of paper and online survey respondents and their results. Using LimeSurvey, we constructed an online survey environment that enabled responses via computer or mobile device. Respondents could choose whether to respond on paper or online. The response rate for the 2016 survey was 51.8%, an increase of 3.5% over the previous year. Of these responses, 15.8% were made online. Online respondents were mostly primiparous. Further, while there was no difference in the percentage of respondents who provided free responses, the amount written was higher in paper surveys than in online surveys. The combination of paper and online surveys increased convenience for respondents and contributed to improved response rates. In addition, paper surveys were superior in terms of allowing respondents to express their feelings and opinions.

## 1. Introduction

On 11 March 2011, Fukushima Prefecture, Japan sustained major damage both from the Great East Japan Earthquake and Tsunami and the resulting Fukushima Dai-Ichi Nuclear Power Plant Accident. The plumes of radiation released by the explosion at the nuclear power plant were carried northward by wind and rain toward the most densely populated area of Fukushima Prefecture. Shortly after the disaster, the Fukushima Prefecture government created the Fukushima Health Management Survey to assess the long-term effect of low-dose radiation exposure resulting from the accident [1].

Even without direct exposure, disasters generally affect a range of reproductive and birth outcomes and can also lead to an increased risk of harmful situations, such as unplanned pregnancies or sexually transmitted infections [2]. Terrorist attacks, nuclear accidents, natural disasters such as hurricanes, and other types of disasters do not have a direct causal relationship with perinatal outcomes, but they have been shown to potentially affect fetal growth [3,4,5,6,7].

With a mandate from the Fukushima Prefectural government, as part of a series of continuous health management efforts for residents of the prefecture, the Fukushima Medical University has conducted the Pregnancy and Birth Survey of the Fukushima Health Management Survey (hereafter, “the Pregnancy and Birth Survey”) every year since 2011 [3]. The Pregnancy and Birth Survey, which targets pregnant women who have been issued a Maternal and Child Health Handbook, aims to better understand the physical and mental health status of mothers and their children in Fukushima Prefecture, alleviate anxiety experienced by mothers in the region, provide necessary perinatal medical care, and bolster health promotion measures. The Maternal and Child Handbook is provided to every pregnant woman in Japan when they register their pregnancy at their municipal office, which is required under the Maternal and Child Health Act in order to receive maternity benefits; the handbook is used by physicians and midwives to record medical information at every antenatal and postnatal visit.

From 2011 to 2015, the Pregnancy and Birth Survey was either mailed to pregnant women or distributed to them at an obstetrics and gynecology facility during a visit. In the event that a respondent replied to the questionnaire multiple times, we deleted their duplicate answers based on ID information. The survey, a self-administered questionnaire survey which, at that time could be completed on paper only, asked about the following: 1. Mental health status, 2. Current living situation (living as an evacuee or apart from family), 3. Delivery situation and the woman’s health status during pregnancy, 4. Confidence regarding childcare, and 5. Attitude towards future pregnancy. The total number of Pregnancy and Birth Survey subjects and respondents in previous years were as follows: in 2011, 9316 respondents out of 16,001 subjects (response rate: 58.2%); in 2012, 7,181 respondents out of 14,516 subjects (49.5%); in 2013, 7260 respondents out of 15,125 subjects (47.7%); in 2014, 7,132 respondents out of 15,125 subjects (47.2%); and in 2015, 7,031 respondents out of 14,572 subjects (48.3%). Thus, the response rate decreased significantly from 2011 to 2012 but remained nearly constant from 2012 to 2015 (Figure 1).

Since 2016, in addition to the existing paper survey, the Pregnancy and Birth Survey has also been available online via smartphones, tablets, and computers. Due to the recent spread of smartphones and tablets, surveys are now conducted online not only for market research [8,9,10], but also for medical research [11]. However, the adoption of online surveys for collecting statistical data by central and local governments has been rather slow in Japan. Although the proportion of central government surveys that can be taken online increased from about 50% in 2011 to about 80% in 2016, only 37% of respondents chose to take the survey online [12,13]. Given this situation, the present study had the following aims: to determine whether making the survey available online improves response rates, and to identify the characteristics of the paper and online survey respondents. The ultimate aim of the study was to use these results as a basis for creating a more user-friendly interface and to establish an online survey system that makes responding more convenient. 

Because expectant and nursing mothers tend to be younger, and therefore tend to be more likely to be smartphone users, an improved response rate was expected. Apart from the respondents’ characteristics, changes in the way questions are presented may lead to changes in the content of responses (“measurement error”) [14]. A recent small-scale report in Japan comparing online and paper surveys found that answers on self-efficacy and problematic internet use differed even within the same individual [15]. Given concerns that responses would differ according to the survey mode (online versus paper), this study also aimed to verify whether this was in fact the case. 

## 2. Materials and Methods

### 2.1. Subjects

There were 14,154 total subjects included in the 2016 Pregnancy and Birth Survey. For inclusion, women had to fulfill one of the following two conditions: (1) issuance of a Maternal and Child Health Handbook in any municipality in Fukushima Prefecture between August 2015 and 31 July 2016; (2) issuance of a Maternal and Child Health Handbook in a municipality outside Fukushima Prefecture between August 2015 and 31 July 2016 but relocation to Fukushima Prefecture to undergo an obstetric examination and give birth.

### 2.2. Survey Methods

Subjects residing in Fukushima Prefecture were mailed a Notice on the Questionnaire for Pregnancy and Birth Survey, an Online Response Instruction Form, a questionnaire booklet, and a Health Administration Survey for Prefectural Residents (“Questionnaire for Pregnancy and Birth Survey”) from the Fukushima Medical University Radiation Medical Science Center. Subjects who relocated or returned to Fukushima Prefecture were given a survey form when they visited an obstetrics and gynecology facility. All subjects could choose whether to take the survey on paper or online. The online survey was available from the day the first paper survey was mailed on 21 November 2016 until 31 August 2017. Data collection continued until 15 December 2017.

Including the initial mailing, the questionnaire was mailed a total of three times. Subjects were grouped depending on expected delivery dates and surveyed first on November 21 2016; next on 20 January 2017; and then finally on 15 March 2017. Subjects who did not respond were also mailed a reminder on 27 March 2017 and were re-sent the questionnaire booklet on 26 June 2017.

The online survey system’s functional requirements were as follows. Responses could be given on any device which supported a web browser. The display and operability of the system were optimized for computers, smartphones, and tablets. To ensure the system could run without problems on smartphones and tablets, compatibility was tested with the latest version of browsers available at the time of service provision, that is, on Safari for iOS and on Chrome and other browsers for Android. On computers, system compatibility was tested on Internet Explorer versions 10 and later, Firefox, and Chrome.

The online response system was comprised of a login function, a survey response function, and a submission function. Using the open-source online survey system LimeSurvey [16], we customized the platform as follows. (1) We developed a system to merge personal and response information in an environment not connected to the internet. (2) To prevent security issues and to protect responses from identity theft, we developed a function for changing the default password. This involved asking survey subjects to log in with the ID issued to them, after which they were then permitted to change their password to one of their own choosing. (3) The input form’s screen structure was identical for computers, smartphones, and tablets. (4) The following check functions were available: a skipped input check, an input range check, a required field check, a logic check for response content between questions, a logic check for response content within the same question, and an alarm display. (5) The following response screen displays were also available: a progress bar indicating current response progress, and pop-up displays explaining the meanings of words in questions.

### 2.3. Analysis Methods

For the 2016 survey, responses were obtained from 7326 women for an overall response rate of 51.8%. After excluding surveys with missing or invalid responses, surveys not submitted within the survey period, and surveys from respondents outside of Fukushima Prefecture, the remaining 7163 respondents (overall response rate: 50.6%) were analyzed as part of this study. Subjects were divided into a paper response group and an online response group, and response rates were examined for both groups. The following characteristics of the respondents were then analyzed: by age, area of residence, and parity. The main survey outcomes analyzed included depressive tendency, preterm birth, low birth weight, and deformities in children. In addition, presence/absence of free responses, categorized content (i.e., economic anxiety, physical or mental disorders, desire for childcare services, childcare consultations), and length of free responses, were analyzed. The odds ratio for online response was calculated using a logistic regression analysis. Area of residence, parity, and the main survey outcomes of depressive tendency, preterm birth, low birth weight, and deformities in children were all adjusted for age. Response data was tabulated and analyzed using IBM SPSS Statistics 21.0. (IBM, Armonk, NY, USA).

### 2.4. Ethical Considerations

The present study was conducted with the approval of the Fukushima Medical University Institutional Review Board (approval nos. 1317 and 2333). The objective of the survey was written on the instructions included with the paper survey; responding to the survey was construed as consent to participate. The online survey also included instructions on screen; clicking the “Complete Responses” button was construed as consent to participate.

## 3. Results

### 3.1. Response Rates

Of the 7163 subjects analyzed in the 2016 survey, 6029 subjects responded by paper, while 1134 subjects responded online (Table 1). Thus, online responses accounted for 15.8% of all responses.

Comparisons of percentages of respondents by age in 2015 and 2016 (before and after the introduction of the online survey, respectively) were as follows. Respondents aged ≤24 years accounted for 9.7% of respondents in 2015 and 10.1% of respondents in 2016. Respondents aged 25−29 years accounted for 29.9% of respondents in 2015 and 30.7% of respondents in 2016. In the older age groups, the proportion declined in 2016.

In the 2016 survey, the number of paper responses peaked 3−4 days after the questionnaire was mailed, while the number of online responses peaked 1−2 days after mailing (Figure 2). The same trends were observed after reminders were sent via postcard, and the survey form was re-sent. Of the subjects who responded to the survey online, 28.0% of subjects did so by computer, while 72.0% of subjects did so by smartphone or tablet.

### 3.2. Characteristics of Online Respondents

Online respondents demonstrated the following characteristics (Table 1). First, online respondents were significantly younger (31.1 ± 5.0 years) than paper respondents (31.5 ± 5.0 years) (*p* < 0.01). Comparing the percentages of primiparous and multiparous women revealed that the percentage of online responses was significantly higher among primiparous women (19.2%; *p* < 0.001), even after adjusting for age. Additionally, the highest percentage of online respondents came from Hamadori (16.2%), the closest region to the Fukushima Daiichi Nuclear Power Plant; followed by Aizu (15.8%) and Nakadori (15.7%). However, there were no statistically significant differences among these percentages of online responses. The percentage of online respondents also did not differ significantly between those with and without with depressive tendencies (Table 2).

Comparing the presence or absence of free responses did not reveal a significant difference between the paper survey and online survey groups (Table 3). However, free responses were significantly longer among paper respondents (164.3 ± 144.7 characters) compared to online respondents (118.8 ± 105.5 characters) (*p* < 0.001).

## 4. Discussion

In order to increase the Pregnancy and Birth Survey response rate, the survey was made available online beginning in 2016. As a result, the overall response rate increased by 3.5% over the previous year. Although the online survey was made available on computers, smartphones, and tablets, the vast majority of online respondents used smartphones and tablets. A 2017 white paper on information and communications [17] showed that in 2016, smartphones were owned by 94.2% of people in their twenties, 90.4% of people in their thirties, and 79.9% of people in their forties. The subjects of the Pregnancy and Birth Survey, who are generally in their twenties to thirties, appeared to demonstrate a strong affinity for smartphones, which may have led to the improvement in the response rate [18]. Previous studies which substituted or combined online surveys for paper-based ones have shown that such substitution has no significant effect on increasing cost or declining response rate [19,20] and that online surveys can be viewed as either a substitute or a complementary method in the scope of epidemiological modes of data collection [21].

Comparing the response rate by age group in the 2015 and 2016 Pregnancy and Birth Surveys (before and after the introduction of online surveys, respectively), younger subjects, who would likely have no resistance to using a smartphone, had a disproportionate amount of online responses compared to other age groups. Contrary to our expectations, this did not result in the majority of the younger group responding online. Rather, parity was significantly associated with the choice of response method, even when adjusting for age. Primiparous women were more likely to choose the online survey compared with multiparous women, even when adjusting for age. One possible explanation for this result involves the online survey system itself. The online survey was set up so that responses could be temporarily saved if the respondent was not able to complete the survey in one sitting. In the case of not being able to finish in one sitting, the respondent would have to access the survey more than once to complete it. Studies have shown that primiparous women tend to have more free time and will more proactively accept childcare services than multiparous women [22] and thus may have been able to take enough time to complete the survey in one sitting or access it multiple times until completion.

A shorter, online follow-up survey of just seven questions was also conducted four years after birth. As of 30 June 2018, the follow-up to the 2013 survey had an online response rate of 23.9%, which was higher than the 15.7% online response rate to the present survey. A previous study compared paper surveys alone with mixed-mode surveys (those which combined paper and online surveys) and found that responses did not differ between the two types of surveys. It also concluded that mixed-mode surveys were superior for a population of young, highly-educated subjects [23]. In order to improve the online response rate of mothers for whom childcare prevents setting aside a long, stable period of time will require improving online response methods. These methods could include creating a shorter questionnaire that can be completed more quickly, or enabling easier re-access to the online survey after responses are temporarily saved.

While the subjects’ characteristics in the present study differed according to the method by which they responded to the survey, the primary pregnancy outcomes that the survey focused on did not show any differences based on response method. This was consistent with the results obtained in a study by Seward et al. [21]. Confirming the limited effect of the online survey on the main results was an assuring finding for data analysis and presentation. In other words, measurement error was negligible in the present study.

Interestingly, free responses were significantly longer in the paper survey results compared with the online surveys, conceivably because paper surveys allow respondents to communicate their feelings and opinions in greater detail. While responding online is easy, paper surveys may be more suitable for subjects who wish to convey their feelings in writing. Therefore, the use of paper surveys versus online surveys may need to be tailored to the individual subject. It is also important that paper surveys be retained as an option for subjects who wish to communicate how they feel at that point in time.

The present study was limited in that, although it demonstrated that combining paper surveys with online surveys yields a great benefit, there was no examination of disadvantages in terms of cost. The study also offered no solutions whatsoever to issues inherent to online surveys, such as protection of personal information and security. In fact, some respondents voiced concerns about data security. Furthermore, we lacked data on educational level and familiarity with internet use that could potentially influence respondents’ selection of survey mode (online versus paper). However, since the main purpose of the Pregnancy and Birth Survey was to monitor mothers’ health, a question about internet use was beyond the survey’s scope. Going forward, we will strive to continue improving response rates by analyzing response content in further detail and improving the user interface.

## 5. Conclusions

The present study demonstrated the following: 1. Providing subjects with the option to take surveys either on paper or online, in accordance with their individual characteristics, led to improved overall response rate; 2. Response behaviors differed based on subjects’ basic characteristics such as parity, age, and region; 3. Differences in response method were not associated with the main results (mother’s depressive tendency, preterm birth, child’s low birth weight/deformities); 4. Free responses were longer in paper surveys than in online surveys, indicating that paper surveys are superior for allowing respondents to express their feelings and opinions more fully. Thus, the combination of paper and online surveys makes surveys more convenient for respondents, thereby contributing to an improvement in response rates. Many reports have discussed online versus paper-based response rates, but in this study, it appeared that respondents chose the paper or the online survey depending on what issues they wished to raise, indicating the effectiveness of using a dual-survey method.

## Figures and Tables

**Figure 1 ijerph-16-01946-f001:**
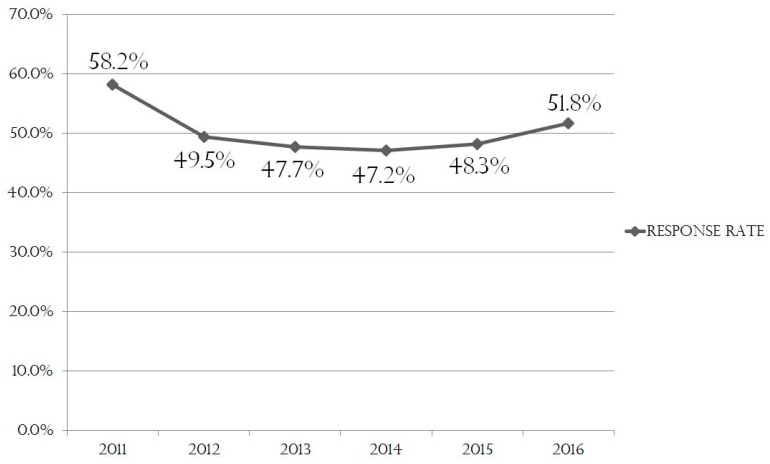
Overall response rate by year.

**Figure 2 ijerph-16-01946-f002:**
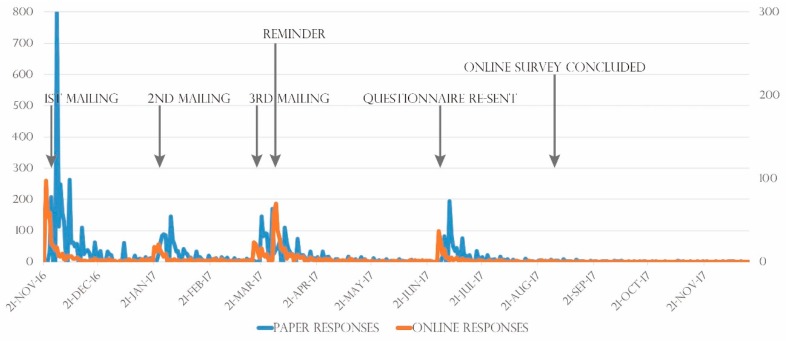
Daily numbers of respondents by response method.

**Table 1 ijerph-16-01946-t001:** Characteristics of Pregnancy and Birth Survey 2016 subjects: comparisons by survey method.

	Paper Responses (*n* = 6029)	Online Responses (*n* = 1134)	AOR	95% CI	*p* Value *
Maternal age					
≤24 years	497 (83.5%)	98 (16.5%)	1.00		
25−29 years	1619 (82.2%)	350 (17.8%)	1.10	0.86–1.40	0.46
30−34 years	2175 (84.8%)	390 (15.2%)	0.91	0.71–1.16	0.44
≥35 years	1738 (85.4%)	296 (14.6%)	0.86	0.67–1.11	0.25
Area of residence (3 regions of Fukushima)					
Nakadori (*n* = 4496)	3790 (84.3%)	706 (15.7%)	1.00		
Hamadori (*n* = 1700)	1425 (83.8%)	275 (16.2%)	1.03	0.89–1.20	0.68
Aizu (*n* = 967)	814 (84.2%)	153 (15.8%)	1.01	0.83–1.22	0.93
Parity					
Primiparous	2663 (80.8%)	632 (19.2%)	1.00		
Multiparous	3155 (86.4%)	495 (13.6%)	0.68	0.59–0.77	<0.001
Singleton/Twin					
Singleton	5975 (84.2%)	1,125 (15.8%)	1.00		
Twin	52 (85.2%)	9 (14.8%)	0.94	0.46–1.92	0.87

* Logistic regression model was used to adjust for age (for items other than age).

**Table 2 ijerph-16-01946-t002:** Pregnancy outcomes: comparisons by survey method.

	Paper Responses (*n* = 6029)	Online Responses (*n* = 1134)	*p* Value *
Pregnancy outcome			
Birth	5958 (98.8%)	1121 (98.9%)	
Other	71 (1.2%)	13 (1.1%)	1.00
Depressive tendency			
No	4732 (79.0%)	896 (79.2%)	
Yes	1261 (21.0%)	236 (20.8%)	0.76
Preterm birth			
No	5817 (97.0%)	1088 (96.9%)	
Yes	180 (3.0%)	35 (3.1%)	0.68
Low birth weight			
No	5415 (90.9%)	1017 (91.0%)	
Yes	544 (9.1%)	101 (9.0%)	0.97
Congenital defects/abnormalities			
No	5748 (97.4%)	1093 (97.7%)	
Yes	156 (2.6%)	26 (2.3%)	0.63

* Logistic regression model was used to adjust for age.

**Table 3 ijerph-16-01946-t003:** Free responses: comparisons by survey method.

	Paper Responses (*n* = 6029)	Online Responses (*n* = 1134)	*p* Value *
Free response			
No	791 (13.1%)	158 (13.9%)	
Yes	5238 (86.9%)	976 (86.1%)	0.47
Categorized content			
Economic anxiety			
No	5955 (98.8%)	1120 (98.8%)	
Yes	74 (1.2%)	14 (1.2%)	1.00
Physical or mental disorders			
No	5841 (96.9%)	1113 (98.1%)	
Yes	188 (3.1%)	21 (1.9%)	<0.05
Desire for childcare services			
No	5806 (96.3%)	1099 (96.9%)	
Yes	223 (3.7%)	35 (3.1%)	0.34
Childcare consultations			
No	5799 (96.2%)	1111 (98.0%)	
Yes	230 (3.8%)	23 (2.0%)	<0.01
	164.3 ± 144.7	118.8 ± 105.5	<0.001

* Fisher’s exact test or *t*-test was used.

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
