# Peer review of "Development and Implementation of an Internet Survey to Assess Community Health in the Face of a Health Crisis: Data from the Pregnancy and Birth Survey of the Fukushima Health Management Survey, 2016"

_ijerph, 2019, doi:10.3390/ijerph16111946_

Round 1

Reviewer 1 Report

In this manuscript the authors evaluate the use of an internet-based survey compared to paper responses for collection of community health data in the Fukushima region of Japan.  The paper is reasonably well written, and then analysis performed is appropriate.

The bigger question is why this is relevant.  The analysis is definitely appropriate for the agency and/or researchers conducting and administering the survey, but there is little of broader scientific interest.  The paper does not contribute to advancing survey research methods, and the authors don't connect their findings to larger current methodological discussions.  

There are some minor typographical errors, for example line 91 'fulfil' instead of 'fulfill'.

This reviewer would have been much more interested to learn about pregnancy outcomes in this sample, and hopes that will be subject of a forthcoming report.

Author Response

Reviewer 1

Comments and Suggestions for Authors

In this manuscript the authors evaluate the use of an internet-based survey compared to paper responses for collection of community health data in the Fukushima region of Japan.  The paper is reasonably well written, and then analysis performed is appropriate.

The bigger question is why this is relevant.  The analysis is definitely appropriate for the agency and/or researchers conducting and administering the survey, but there is little of broader scientific interest.  The paper does not contribute to advancing survey research methods, and the authors don't connect their findings to larger current methodological discussions.

RESPONSE:

The adoption of online surveys has been rather slow in Japan, and only about half of government surveys were taken online in 2011 (the year of the Fukushima nuclear power plant accident) (reference 1). The proportion of such surveys has increased to about 80% as of 2018, and the government survey webpage was updated in 2016 to accept access from smartphones and tablets. Despite these developments, only 37% of respondents selected the online mode of the 2015 Census (reference 2). Therefore, our trial evaluating the adoption of online surveys in a prefectural survey could provide new insights for wider application. The following explanations have been added in the Introduction.

Introduction, L(78)-(82): However, the adoption of online surveys for collecting statistical data by central and local governments has been rather slow in Japan. Although the proportion of central government surveys that can be taken online increased from about 50% in 2011 to about 80% in 2016, only 37% of respondents chose to take the survey online [12, 13]. Given this situation, the present study had the following aims…

Introduction, L(90)-(94): Apart from the respondents’ characteristics, changes in the way questions are presented may lead to changes in the content of responses (“measurement error”) [14]. A recent small-scale report in Japan comparing online and paper surveys found that answers on self-efficay and problematic internet use differed even within the same individual [15]. Given concerns that responses would differ according to the survey mode (online versus paper), this study also aimed to verify whether this was in fact the case.

There are some minor typographical errors, for example line 91 'fulfil' instead of 'fulfill'.

RESPONSE:

The manuscript was again professionally edited by a native speaker familiar with this area of research.

This reviewer would have been much more interested to learn about pregnancy outcomes in this sample, and hopes that will be subject of a forthcoming report.

RESPONSE:

Thank you for your comment. We have more clearly stated that the present paper is rather technical, with an aim to investigate effects of introducing the online survey. In addition, pregnancy outcomes between the two groups (paper and online survey respondents) were shown to be comparable (Table 2).

Reviewer 2 Report

Dear author,

I have  read your response. I thought you had answered my some questions. However some questions still exist. I hope you can mark the line and page of your change next times.

1.  To my knowledge, whether or not to accept the online survey depends mainly on age, educational level, mastery of electronic products and familiarity with the Internet use. It may associate to the strategy whether you encourage the respondents to fill in the survey online. In general, research contents should be carefully considered so that the goals and results of the study are clear to the reader. If the authors did not consider such factors, it may bring the so-called omitted variable which named the endogenous problem. Because of the missing of true influence variables missing, for most of the variables in the manuscript, results of logic analyse was not significant.

2.  Of introduction part, you need to be more precise how you are building the extant theoretical literature for your specific research question. It seems that the authors’ emphasis is the description of the data. However, there is a lack of literature basis of this research.

3. Of discussion part, negative outcomes were not well discussed. The content of table 2 was not discussed.

4. Line 208 to 210, “Contrary to our expectation, this did not result in the majority of the younger group responding online. Rather, parity was significantly associated with the choice of response method even when adjusting for age.”

These words were not linking to the results of the above text. In table 1, there is no results indicating the results “adjusting for age”.

Kind regards,

Jianqian Chao

Author Response

Reviewer 2

Comments and Suggestions for Authors

I have read your response. I thought you had answered my some questions. However some questions still exist. I hope you can mark the line and page of your change next times.

RESPONSE:

We have highlighted changes in the previous (yellow) and present (blue) revisions.

1. To my knowledge, whether or not to accept the online survey depends mainly on age, educational level, mastery of electronic products and familiarity with the Internet use. It may associate to the strategy whether you encourage the respondents to fill in the survey online. In general, research contents should be carefully considered so that the goals and results of the study are clear to the reader. If the authors did not consider such factors, it may bring the so-called omitted variable which named the endogenous problem. Because of the missing of true influence variables missing, for most of the variables in the manuscript, results of logic analyse was not significant.

RESPONSE:

Thank you for your insightful comment. We have added the lack of data on education and familiarity with internet use as one of our study limitations.

Discussion, L(256)-(259): Furthermore, we lacked data on educational level and familiarity with internet use that could potentially influence respondents’ selection of survey mode (online versus paper). However, since the main purpose of the Pregnancy and Birth Survey was to monitor mothers’ health, a question about internet use was beyond the survey scope.

2. Of introduction part, you need to be more precise how you are building the extant theoretical literature for your specific research question. It seems that the authors’ emphasis is the description of the data. However, there is a lack of literature basis of this research.

RESPONSE:

We have expanded the explanation of our study aims by citing previous research on survey methods.

Introduction, L(78)-(82): However, the adoption of online surveys for collecting statistical data by central and local governments has been rather slow in Japan. Although the proportion of central government surveys that can be taken online increased from about 50% in 2011 to about 80% in 2016, only 37% of respondents chose to take the survey online [12, 13]. Given this situation, the present study had the following aims…

Introduction, L(90)-(94): Apart from the respondents’ characteristics, changes in the way questions are presented may lead to changes in the content of responses (“measurement error”) [14]. A recent small-scale report in Japan comparing online and paper surveys found that answers on self-efficay and problematic internet use differed even within the same individual [15]. Given concerns that responses would differ according to the survey mode (online versus paper), this study also aimed to verify whether this was in fact the case.

3. Of discussion part, negative outcomes were not well discussed. The content of table 2 was not discussed.

RESPONSE:

The text in the Discussion section concerning negative outcomes was slightly expanded to reflect what was added in the study aims.

Introduction, L(90)-(94): Apart from the respondents’ characteristics, changes in the way questions are presented may lead to changes in the content of responses (“measurement error”) [14].... Given concerns that responses would differ according to the survey mode (online versus paper), this study also aimed to verify whether this was in fact the case.

Discussion, L(243)-(244): In other words, measurement error was negligible in the present study.

4. Line 208 to 210, “Contrary to our expectation, this did not result in the majority of the younger group responding online. Rather, parity was significantly associated with the choice of response method even when adjusting for age.”

These words were not linking to the results of the above text. In table 1, there is no results indicating the results “adjusting for age”.

RESPONSE:

The OR in Table 1 was corrected as AOR. In addition, the phrase “even after adjusting for age” (L(183)) was added in text in the Results.

Reviewer 3 Report

I only made a few suggestions the first time and they don't appear to have been adopted.   For example, I don't think the difference in maternal age is notable, even though it is statistically significant.  

I don't see the figures.  It isn't clear if they were changed in response to the reviews.   

Author Response

Reviewer 3

Comments and Suggestions for Authors

I only made a few suggestions the first time and they don't appear to have been adopted.   For example, I don't think the difference in maternal age is notable, even though it is statistically significant.

RESPONSE: 

We agree that a slight difference in age, even if statistically significant, does not have a clinical implication. In the previous revision, we categorized age, and shifted our focus more toward the difference in parity.

I don't see the figures. It isn't clear if they were changed in response to the reviews.  

RESPONSE:

The figures have been added at the end of the revised version.

Round 2

Reviewer 2 Report

It is ok!

This manuscript is a resubmission of an earlier submission. The following is a list of the peer review reports and author responses from that submission.

Round 1

Reviewer 1 Report

The intent of this manuscript is to describe the development of a database of environmental exposures of a potentially carcinogenic nature for the state of Oklahoma, USA.  The body of the manuscript is very succinct (the length of the text is only 6 double-spaced pages) but not well integrated with research literature.  There is also one figure, consisting of four panels showing example exposure maps, and a supplemental table of more than 40 text pages.

Its quite unclear from reading this manuscript why anyone else would want to read it.  If the authors' intent is to provide a framework for the development of spatiotemporal exposure databases, then write a paper about how to do that, describing the goals and purpose, steps involved in identifying potential data, the data structure for storing these data and retrieving specific queries, how to evaluate data quality, and how to maintain the database into the future.

If the intent is to try to get another publication out of an Oklahoma-based cancer epidemiology project, this paper doesn't rise to the bar.  There's nothing in the methods about how various data sources were evaluated, how different scales or inconsistent measurements over time were addressed, how remotely sensed data was handled compared to data from environmental monitors, how a user might inter-collate user-obtained data with data from the database, whether there are spatial or spatio-temporal tools integrated into the database.  In fact, it appears that the entire database is managed within ArcGIS (p 4), which does not have a structure sufficient to handle the complexity of data that should be in this repository.

The authors also don't go into sufficient detail concerning what content to include in the database, mentioning some sources from the National Toxicology Program but not addressing the vast environmental epidemiology literature.  For that matter, why only consider exposures that might be associated with cancer - there are many other chronic and neuro-degenerative disorders that might have an environmental etiology.

All in all, this manuscript seems to be very early in the process, and requires a great deal of elaboration to become a worthwhile addition to the peer-reviewed literature.

Reviewer 2 Report

This manuscript describes the response rate for two methods of administering a survey (paper vs. online) in the Fukushima Prefecture.   The description of the survey itself and its purpose of the are very interesting, although that has been described in a number of other manuscripts.     

The intent to improve response rates using an electronic delivery method is laudable.   Some significant differences between the groups that responded were noted. However, the differences may not be clinically important.   For example, the difference in ages of the two groups is statistically significant, yet the average age in both groups is approximately 31 years.

A couple of minor comments:

Figure 1 is not necessary.  Redundant with figure 2. 

 The Issuance of a Maternal and Child Health Handbook – it isn’t clear what this is and when one receives it.   Is it distributed when a woman first sees a provider after becoming pregnant?    

The manuscript is concise, and the methods are well-described. However, I do not believe that the findings in this report are particularly striking, nor are they generalizable to others.   I commend the authors for doing a thorough analysis of the data.   

Reviewer 3 Report

Dear author,

First of all, I would like to congratulate you with the completion of your extensive study. I think you’ve invested huge efforts into preparing and conducting this study. In general I find your study important and interesting. I hope my comments below will help you to improve the manuscript.

1.       Originality/Novelty: Is the question original and well defined? Do the results provide an advance in current knowledge?

This manuscript reports on the survey available online improved response rates and the characteristics of paper and online survey respondents and their results. The results provide an advance in current knowledge.

2.  Significance: Are the results interpreted appropriately? Are they significant? Are all conclusions justified and supported by the results? Are hypotheses and speculations carefully identified as such?

Many reports have discussed outcomes such as spontaneous abortion, preterm delivery,

congenital anomalies, low birth weight, depression, and anxiety, and these outcomes are solid to the respondents. To my knowledge, whether or not to accept the online survey depends mainly on age, educational level, mastery of electronic products and familiarity with the Internet use. It may associate to the strategy whether you encourage the respondents to fill in the survey online. In general, research contents should be carefully considered so that the goals and results of the study are clear to the reader.

3. Quality of Presentation: Is the article written in an appropriate way? Are the data and analyses presented appropriately? Are the highest standards for presentation of the results used?

The authors have reported twice on the results of Pregnancy outcome, Depressive tendency,

Preterm birth, Low birth weight, Congenital defects/abnormalities in Table 1 and Table 2. Moreover, there is no explanation of Table 2 in the context.

4. Scientific Soundness: is the study correctly designed and technically sound? Are the analyses performed with the highest technical standards? Are the data robust enough to draw the conclusions? Are the methods, tools, software, and reagents described with sufficient details to allow another researcher to reproduce the results?

   On line 56, “the Pregnancy and Birth Survey was either mailed to pregnant women or was distributed to them at an obstetrics and gynecology facility during a visit”. It is not clear that if the questionnaire registered with identified information to avoid filling repeatedly.

In 2006, the authors reported 7326 respondents (Fig 2). However, the authors didn’t report

whether all of these questionnaires were valid. The total number of respondents in 3 regions of Fukushima is 7163 (Table 1). It is suggested to make the criterion of the valid questionnaires clear.

On line 186, the authors have cited two previous research that illustrated there were minor

differences between online and paper questionnaire on responding rates or cost. However, these two references became less value for the rapid development of internet. The reference below has shown some new evidence:

Seward, R. J., et al. 2018. Psychometric Properties and Norms for the Strengths and Difficulties Questionnaire Administered Online in an Australian Sample. Australian Psychologist, 53(2), 116-24.
5. Interest to the Readers: Are the conclusions interesting for the readership of the Journal? Will the paper attract a wide readership, or be of interest only to a limited number of people? (please see the Aims and Scope of the journal)

Some editing must be done.

Overall Merit: Is there an overall benefit to publishing this work? Does the work provide an advance towards the current knowledge? Do the authors have addressed an important long-standing question with smart experiments?

Some editing must be done.

6.       English Level: Is the English language appropriate and understandable?

Yes.